# Reconstructing Binary Signals from Local Histograms

**DOI:** 10.3390/e24030433

**Published:** 2022-03-21

**Authors:** Jon Sporring, Sune Darkner

**Affiliations:** 1Department of Computer Science, University of Copenhagen, 2100 Copenhagen, Denmark; darkner@di.ku.dk; 2Center for Quantifying Images, MAX IV (QIM), 2800 Kgs. Lyngby, Denmark

**Keywords:** local histograms, metameric classes, reconstruction

## Abstract

In this paper, we considered the representation power of local overlapping histograms for discrete binary signals. We give an algorithm that is linear in signal size and factorial in window size for producing the set of signals, which share a sequence of densely overlapping histograms, and we state the values for the sizes of the number of unique signals for a given set of histograms, as well as give bounds on the number of metameric classes, where a metameric class is a set of signals larger than one, which has the same set of densely overlapping histograms.

## 1. Introduction

A histogram is a central tool for analyzing the content of signals while disregarding positional relations. It is useful for tasks such as setting thresholds for detecting extremal events and for designing codes in communication tasks. In [1], the three fundamental scales for histograms for discrete signals (and images) were presented: the intensity resolution or the bin-width, the spatial resolution, and the local extent, for which a histogram is evaluated. Even when fixing these scale parameters, it is still essential to consider the sampling phase, since in general, we do not know the location of the interesting signal parts, and thus, we must consider all phases, or equivalently, all overlapping histograms and all histograms for different positions of the first left bin-edge.

A natural question, when using local histograms for signals and image analysis, is: How many signals share a given set of overlapping, local histograms (illustrated in Figure 1)? Out of pure theoretical interest, in this paper, we took a first step in answering this question by considering densely overlapping histograms of binary signals.

As an example, consider the signal [0;0;1]. Its global histogram is (2,1), i.e., there are two “0” values and one “1” value.
(1)$$\left[0;0;1\right]\overset{\mathrm{histogram}}{\to }\left(2,1\right).$$ Given the histogram, the only possible signals are [0;0;1], [0;1;0], [1;0;0], i.e.,
(2)$$\left(2,1\right)\overset{\mathrm{histogram}}{\leftarrow }\left\{\begin{array}{c} {}[0;0;1],\\ {}[0;1;0],,\\ {}[1;0;0],. \end{array}\right.$$ This is a much smaller number than $2^{3}$ possible signals, but it is not a bijective relation. The representation power of the histogram may be quantified as the conditional entropy of signals given their histogram. For binary signals of length three, the histogram of a signal may be summarized by its count of “1”-values, since the number of “0”-values will be three minus this count. For length three binary signals, there are eight different signals [0;0;0],[0;0;1],…[1;1;1], which have four different histograms where the counts of “1” values are 0, 1, 2, and 3, respectively, and the corresponding number of signals counted by their “1”-values is 1, 3, 3, and 1. Given a histogram, the conditional probability of each of these corresponding signals is thus $0,\log_{2}3,\log_{2}3,0$, and the conditional entropy may thus be found to be approximately 1.2 bit.

In this paper, we did not focus on coding schemes for signals, but on the expression power of local overlapping histograms. Thus, consider again the signal in (Equation 1), but now with a set of local histograms of extent two, in which case, the histograms are calculated for the overlapping sub-signals,
(3)$$\left[0;0;1\right]\overset{\mathrm{sliding}\;\mathrm{window}}{\to }\left\{\begin{array}{c} \left[0;0\right]\overset{\mathrm{histogram}}{\to }\left(2,0\right)\\ \left[0;1\right]\overset{\mathrm{histogram}}{\to }\left(1,1\right). \end{array}\right.$$ In this case, there is only one signal that has this sequence of overlapping histograms, since, by the first histogram, we know that the first two values are “0”, and in combination with the second histogram, we conclude that the last value must be “1”; thus, the signal must be [0;0;1]. In contrast, the signals, [0;1;0] and [1;0;1] have the same local histograms,
(4)$$\left[0;1;0\right]\overset{\mathrm{sliding}\;\mathrm{window}}{\to }\left\{\begin{array}{c} \left[0;1\right]\overset{\mathrm{histogram}}{\to }\left(1,1\right)\\ \left[1;0\right]\overset{\mathrm{histogram}}{\to }\left(1,1\right), \end{array}\right.$$
and:(5)$$\left[1;0;1\right]\overset{\mathrm{sliding}\;\mathrm{window}}{\to }\left\{\begin{array}{c} \left[1;0\right]\overset{\mathrm{histogram}}{\to }\left(1,1\right)\\ \left[0;1\right]\overset{\mathrm{histogram}}{\to }\left(1,1\right), \end{array}\right.$$
in which case the signal↔histograms is not a bijective relation. However, the local histogram and the global histogram together uniquely identify the signals. As these examples show, the relation between local histograms and signals is non-trivial, and in this paper, we considered the space of possible overlapping local histograms and the number of signals sharing a given set of local histograms.

In the early 20th Century, much attention was given to the lossless reconstruction of signals, in particular using error-correction codes, where the original signal is sent together with added information [2]. Such additional information could be related to the histogram of the original signal. Later, the reconstruction of signals and images became more pressing problems, primarily as a way to compress images without losing essential content, and this resulted in still widely used image representation standards such as mpeg, tiff, and jpeg.

While signal representation has still been of some concern in the 21st Century, advances in hardware means that more attention has been given to image representation and, in particular, to qualifying the information content of image features. In [3], the authors introduced the concept of metameric classes for local features. The authors considered scale-space features and investigated the space of images that share these features. They further presented several algorithms for picking a single reconstruction. An extension of this approach was presented in [4], where patches at interest points of an original image were matched with patches from a database by a feature descriptor such as SIFT [5]. The database patches were then warped and stitched to form an approximation of the original image. In [6], the authors presented a reconstruction algorithm based on binarized local Gaussian weighted averages and using convex optimization. The theoretical properties of the reconstruction algorithm is still an open research question. In [7], images were reconstructed from a histogram of a densely sampled dictionary of local image descriptors (bag-of-visual-words) as a jigsaw puzzle with overlaps. They showed that their method resulted in a quadratic assignment problem and used heuristics to find a good reconstruction. In [8], the authors investigated the reconstruction of images from a simplified SIFT transform. The reconstruction was performed based on the SIFT-key points and their discretized local histograms of the gradient orientations, and several models were presented for choosing a single reconstruction from the possible candidates. In [9], a convolutional neural network was presented that reconstructs images from a regularly, but sparsely sampled set of image descriptors. The network was able to learn image priors and was able to reconstruct images from both classical features such as SIFT and representations found in AlexNET [10]. This was later extended in [11], where an adversarial network was investigated for reconstruction from local SIFT features.

Our work is closely related to [12], which discussed the relation between FRAME [13] and Julesz’s model for human perception of textures [14]. In [13], the authors defined a Julesz ensemble as a set of images that share identical values of basic features statistics. Although not considered in their works, histogram bin-values can be considered a feature statistics, and hence, the metameric classes presented in this paper are Julesz ensembles in the sense of [13]. In [12], they considered normalized histograms of images filtered with Gabor kernels [15], and they considered the limit of the spatial sampling domain converging to $Z^{2}$. Their perspective may be generalized to local histograms; however, their results only hold in the limit.

This paper is organized as follows: First, we define the problem in Section 2. Section 3 describes an algorithm for finding the signal(s) that has (have) a specific set of local histograms. In Section 2, constraints on possible local histograms and the size of metameric classes are discussed, and finally, in Section 5, we present our conclusions.

## 2. Metameric Signal Classes

We were interested in the number of signals that have the same set of local histograms. In case there is more than one, then we call this a metameric signal class (or just a metameric class) defined by their shared set of local histograms. We define signals and their local histograms as follows: Consider an alphabet A={0,1},l=|A|=2 and a one-dimensional signal $S\in A^{n},n>1$, which we denote $S=[s_{0};s_{1};\cdots ;s_{n-1}]$ and where $s_{i}\in A$ is the value of *S* at position *i*. For a given window size 1<m≤n, we considered all local windows $S_{j}=\left[s_{j};s_{j+1};\ldots ;s_{j+m-1}\right],0\leq j\leq n-m$ and their histograms $h_{j}:A\to Z_{+}$,
(6)$$\begin{array}{cc} h_{j} & =(h_{j}\left(0\right),h_{j}\left(1\right)) \end{array}$$
(7)$$\begin{array}{cc} h_{j}\left(a\right) & =\sum_{i=j}^{j+m-1}\delta \left(s_{i}-a\right), \end{array}$$
where δ is the Kronecker delta function, defined as:(8)$$\delta \left(x\right)=\left\{\begin{array}{c} 1,\;\mathrm{when}\;x=0,\\ 0,\;\mathrm{otherwise}. \end{array}\right.$$ All local histograms of for the signal *S* are $H_{S}=\left[h_{0};h_{1};\ldots ;h_{n-m}\right]$.

As an example, consider the signal,
(9)$$\begin{array}{cc} S & =[0;1;1;1;0], \end{array}$$
in which case n=5. For m=3, the windows are:(10)$$\begin{array}{cc} S_{0} & =[0;1;1], \end{array}$$(11)$$\begin{array}{cc} S_{1} & =[1;1;1], \end{array}$$(12)$$\begin{array}{cc} S_{2} & =[1;1;0], \end{array}$$
and the corresponding histograms are:(13)$$\begin{array}{cc} h_{0} & =(1,2), \end{array}$$(14)$$\begin{array}{cc} h_{1} & =(0,3), \end{array}$$(15)$$\begin{array}{cc} h_{2} & =(1,2), \end{array}$$
or equivalently, in short form,
(16)$$\begin{array}{c} H_{S}=\left[\left(1,2\right);\left(0,3\right);\left(1,2\right)\right]. \end{array}$$

In some cases, two different signals will have the same set of histograms, and we call these signals metameric, i.e., they appear identical w.r.t. their histograms. We say that they belong to the same metameric class given by their common histogram sequence. For example, when n=5 and m=2, the signals,
(17)$$\begin{array}{cc} S & =[0;1;0;1;0], \end{array}$$
(18)$$\begin{array}{cc} S^{\prime } & =[1;0;1;0;1], \end{array}$$
have the same sequence of n−m+1=4 histograms,
(19)$$\begin{array}{c} H_{S}=H_{S^{\prime }}=\left[\left(1,1\right);\left(1,1\right);\left(1,1\right);\left(1,1\right)\right], \end{array}$$
and thus, *S* and $S^{\prime }$ belong to the same metameric class denoted by [(1,1);(1,1);(1,1);(1,1)].

We were interested in the ability of local histograms to represent signals. Hence, for a given signal and window sizes, we sought to calculate μ, the number of signals *S*, which are uniquely identified by $H_{S}$ and κ, the number of metameric classes. The values of μ and κ for small values of *n* are shown in Table 1. These values were counted by considering all $2^{n}$ different possible signals, which is an approach only possible for small values of *n*. From the table, we observe that we did not find any combination of signal lengths and window sizes without a metameric class; hence, none of the tested combinations yielded a unique relation between the local histograms and the signal. Further, the number of unique signals μ appeared to grow with n/m, and the number of metameric classes κ appeared to be convex in *m* for a large value of *n*.

## 3. An Algorithm for Reconstructing the Complete Set of Signals from a Sequence of Histograms

We constructed an algorithm for reconstructing the one or more signals, which has or have a given sequence of histograms. It was constructed using the following facts:Fact 1There is a non-empty and finite set of signals of size *m*, which share the same histogram *h*. These can be produced as all the distinct permutations of the following signal:
(20)$$S=[\underset{h(0)}{\underbrace{0;\ldots ;0}};\underset{h(1)}{\underbrace{1;\ldots ;1}}],$$
and the number of distinct signals is given by the binomial coefficient:
(21)h(0)+h(1)h(0)=h(0)+h(1)!h(0)!h(1)!;Fact 2Consider the windows $S_{i-1}$ and $S_{i}$ and their corresponding histograms $h_{i-1}$ and $h_{i}$ for i=0..n−m. If $s_{i-1}=s_{i+m-1}$, the histograms will be identical; otherwise, the histograms will differ by the count of one at $s_{i-1}$ and at $s_{i+m-1}$, and $s_{i+m-1}={\bar{s}}_{i-1}$, where $\bar{\cdot }$ is the Boolean “not” operator;Fact 3From Fact 2, it follows that the histogram of $\left[s_{i};s_{i+1};\ldots ;s_{i+m-2}\right]$ is equal to $h_{i}$, but where $h_{i}\left(s_{i+m-1}\right)$ has been reduced by one. We call this $h_{i}^{\prime }$;Fact 4The difference,
(22)$$d_{i}\left(a\right)=h_{i-1}\left(a\right)-h_{i}^{\prime }\left(a\right)=\left\{\begin{array}{c} 1,\;\mathrm{when}\;a=s_{i-1},\\ 0,\;\mathrm{otherwise}. \end{array}\right.$$As a consequence, for candidate signals $S_{i}^{\prime }$ that have histogram $h_{i}$, but that have the wrong value placed at $s_{i+m-1}$, the difference will have both negative and positive values.

Thus, we constructed the following algorithm:Step 1Produce a candidate set of all the distinct signals of size *m* that have the histogram $h_{n-m}$;Step 2For i=n−m−1 …0 and for each element in the candidate set $S_{i}$:Step 2.1Calculate $d_{i}$.Step 2.2If $d_{i}$ does not have the form of (Equation 22), then discard it;Step 2.3Else, derive $s_{i-1}$ from $d_{i}$, and extend the candidate with this value:

The computational complexity of our algorithm is $O\left(K_{m}\left(n-m\right)\right)$, where $K_{m}$ is the maximum value of (Equation 21), since initially, all signals of $h_{n-m-1}$ must be considered, and this set can only shrink when considering earlier values.

A working code in F# is given at:

https://github.com/sporring/reconstructionFromHistograms/reconstructionFromHistograms.fsx. For the specific version discussed in this paper, see the commit of 24 December 2021, https://github.com/sporring/reconstructionFromHistograms/commit/333506e72fd8bea2132857869aca15b54392aa75 (accessed on 24 December 2021).

The code does not give any output, when run, but running in F#-interactive mode allows the user to inspect the key values after running the program, which are:

signal: int list = [0; 1; 0; 1; 1],

histogramList: Map<int,int> list = [map [(0, 2); (1, 1)]; map [(0, 1); (1, 2)]; map [(0, 1); (1, 2)]],

solutions: int list list = [[0; 0; 1; 1; 0]; [0; 1; 0; 1; 1]].

The signal is a sequence of binary digits, and the histogram sequence is represented as a sequence of maps, where each map-entry is an (intensity, count) pair, i.e., map [(0, 2); (1, 1)] above is equal to the histogram (2,1). Finally, the solutions is represented as a sequence of sequences of binary digits. In this case, there is, as we can see, a metameric class of two signals, which shares a sequence of histograms. We verified that the algorithm is able to correctly reconstruct all the signal considered in Table 1 including all the members of the metameric classes.

## 4. Theoretical Considerations on μ and κ

In the following, we consider classes of histogram sequences and relate them to the number of metameric classes for a given family of signals and their sizes.

As a preliminary fact, note that for a window size *m*, all the histograms must have:(23)$$h_{j}\left(0\right)+h_{j}\left(1\right)=m,$$
since all entries in $s_{i}$ are counted exactly once.

### 4.1. Constant Sequence of Histograms ($h_{0}=h_{j}$)

There are two different constant signals of length *n*: [0;0;…;0] and [1;1;…;1]. All neighborhoods and histograms of the constant signals are identical, and a histogram will have one non-zero element with value *m*. These signals cannot belong to a metameric class, since permuting the position of the values in $S_{0}$ does not give a new signal, and they are trivially unique.

In general signals with constant histogram sequences, $h_{0}=h_{j}$, the signal must be periodic, since the only difference in the histogram count of $h_{j}$ and $h_{j+1}$ is that $h_{j}$ includes $s_{j+m}$ and $h_{j+1}$ does not include $s_{j}$. Hence, for $h_{j}=h_{j+1}$, then $s_{j}=s_{j+m}$. For example, [1;0;1;0;1] is a periodic signal for m=2 with histogram $h_{j}=\left(1,1\right),\;j=0\ldots 3$. Any constant sequence of histograms describes a periodic signal, and for non-constant signals (∀a:h(a)<m) and m>1, these histograms describe a metameric class, since some permutations of $S=[S_{0}|S_{0}|\ldots ]$ will produce new signals without changing the histograms due to periodicity. For any n>m, there are $2^{m}-2$ such periodic binary signals.

### 4.2. Global Histogram (m=n)

For a particular $h=h_{0}$, all permutations of the signal belong to the same metameric class. Thus, the number of metameric class is equal to the number of different histograms with sum *m*, except those for constant signals. This corresponds to picking *m* numbers from *A* where repetition is allowed and order does not matter.

Following the standard derivation of unordered sampling with replacement, we visually rewrite the terms in (Equation 23) with a list of “·’s”, where each “·” represents the count of one for a given bin. For example, for m=3, we may have the histogram (2,1), which implies that 3=2+1=··+·. A different histogram could be (0,3), implying that 3=0+3=+···. Hence, any permutation of three “·’s” and one “+” will in this representation give the sum of three, and the number of permutations is equal to the number of ways we can choose *m* out of m+1 positions. Thus, the number of different ways we can pick histograms is given as the binomial coefficient:(24)m+1m=m+11=(m+1)!1!m!=m+1. Out of these, two histograms stem from the constant signals. The remaining histograms have ∀a:h(a)<m, and each of these histograms defines a metameric class, since there will always be more than one signal with such a histogram by (Equation 21). Hence, the number of different histograms is,
(25)κ(m,m)=m−1. This equation confirms the values in Table 1 where n=m.

### 4.3. Smallest Histogram (m=2)

For the case of m=2, we now show that:(26)κ(n,2)=1. Consider the sequence of histograms $\left[h_{0};h_{1};\cdots \right]$. For $h_{0}$, there are three different histograms corresponding to $2^{2}$ different signals. These signals fall into two classes: $s_{0}=s_{1}$ is trivially solvable; however, for $s_{0}\neq s_{1}$, the values of $s_{0}$ and $s_{1}$ are easily identifiable from the histogram $h_{0}$, but their positions are not. Write $h_{0}=\left[\left(s_{0},1\right);\left(s_{1},1\right)\right]$. Now, consider $h_{1}$. Again, if $s_{1}=s_{2}$, then their values are trivially solvable, and since $s_{1}$ is known, then $s_{0}$ can be deduced from $h_{0}$. Therefore, assume that $s_{1}\neq s_{2}$, and write $h_{1}=\left[\left(s_{1},1\right);\left(s_{2},1\right)\right]$. Now, consider $h_{2}$; as before, if $s_{2}=s_{3}$, then their values are trivially solvable by $h_{2}$; hence, $s_{1}$ can be deduced from $h_{1}$ and $s_{2}$, and in turn, $s_{0}$ can be deduced from $s_{1}$ and $h_{0}$. The general structure of the problem is illustrated in Figure 2, and by induction, we see that only the constant sequence of histograms $h_{j}=\left(1,1\right),\;j=0\ldots n-m-1$ is a metameric class; hence, κ(n,2)=1.

### 4.4. The General Case (n>m>2)

Since the sum of a histogram is *m* (see (Equation 23)) and since the histograms for binary signals only have two bins, we can identify each histogram by:(27)$$\sigma_{j}=h_{j}\left(1\right)=m-h_{j}\left(0\right)=\sum_{i=j}^{j+m-1}s_{i},$$
i.e., as the number of one-values in $S_{j}$. Thus, in the following, we identify $h_{j}$ by $\sigma_{j}$. In the following, we consider consecutive pairs of histograms for signals of varying lengths n>m.

Firstly, consider the case n=3 and m=2 and all possible combinations of histograms $h_{0}$ and $h_{1}$, i.e., $\sigma_{0},\sigma_{1}\in \left\{0,1,2\right\}$. The organization of all the $2^{3}$ signals in terms of $\sigma_{0}$ and $\sigma_{1}$ is shown in Table 2. We call such tables transition tables, and we say that each table cell contains a set of signal pieces. For n=3 and m=2, the table illustrates that there is one metameric class shown in cell $\sigma_{0}=\sigma_{1}=1$, since this table cell contains two elements. This case is also discussed in relation to (Equation 26).

Now, consider the case n=4 and m=2. The transition table for $\left(\sigma_{1},\sigma_{2}\right)$ is identical to Table 2. Further, an element in $\left(\sigma_{0},\sigma_{1}\right)$ is related to an element in $\left(\sigma_{1},\sigma_{2}\right)$ by the arrows in the table. For example, if $\left[s_{0};s_{1};s_{2}\right]=\left[0;0;1\right]$, then $\sigma_{0}=0$ and $\sigma_{1}=1$. If $s_{3}=0$, then $\left[s_{1};s_{2};s_{3}\right]=\left[0;1;0\right]$, $\sigma_{0}=1$, and $\sigma_{1}=1$, while if $s_{3}=1$, then $\left[s_{1};s_{2};s_{3}\right]=\left[0;1;1\right]$, $\sigma_{0}=1$, and $\sigma_{1}=2$.

Transition tables contain zero or more $\left[s_{j};s_{j+1};\ldots ;s_{m}\right]$ elements that have histograms $\sigma_{j}$ and $\sigma_{j+1}$. The tables have a particular structure:Fact 5$\left(\sigma_{j},\sigma_{j+1}\right)$ is a tridiagonal table: Since $\sigma_{j}$ and $\sigma_{j+1}$ only differ by the values $s_{j}$ and $s_{j+m}$, then the differences between $\sigma_{j}$ and $\sigma_{j+1}$ can maximally be one. Hence, the table will have a tridiagonal structure;Fact 6Elements on the main diagonal have $s_{j}=s_{j+m}$: On the diagonal $\sigma_{j+1}=\sigma_{j}$, hence:
(28)$$\begin{array}{cc} \sigma_{j} & =\sum_{i=0}^{m-1}s_{j+i}=s_{j}+\sum_{i=1}^{m-1}s_{j+i} \end{array}$$
(29)$$\begin{array}{cc} \sigma_{j+1} & =\sum_{i=0}^{m-1}s_{j+1+i}=s_{j+m}+\sum_{i=0}^{m-2}s_{j+1+i}=s_{j+m}+\sum_{i=1}^{m-1}s_{j+i}=\sigma_{j} \end{array}$$Thus, $s_{j}=s_{j+m}$;Fact 7Elements on the first diagonal above have $s_{j}=0\wedge s_{j+m}=1$: On the first diagonal above, $\sigma_{j+1}=\sigma_{j}+1$, and thus,
(30)$$\begin{array}{cc} \sigma_{j} & =s_{j}+\sum_{i=1}^{m-1}s_{j+i} \end{array}$$
(31)$$\begin{array}{cc} \sigma_{j+1} & =s_{j+m}+\sum_{i=1}^{m-1}s_{j+i}=\sigma_{j}+1 \end{array}$$Thus, $s_{j}=s_{j+m}-1\Rightarrow s_{j}=0\wedge s_{j+m}=1$;Fact 8Elements the first diagonal below have $s_{j}=1\wedge s_{j+m}=0$: On the first diagonal below, $\sigma_{j+1}=\sigma_{j}-1$, and thus,
(32)$$\begin{array}{cc} \sigma_{j} & =s_{j}+\sum_{i=1}^{m-1}s_{j+i} \end{array}$$
(33)$$\begin{array}{cc} \sigma_{j+1} & =s_{j+m}+\sum_{i=1}^{m-1}s_{j+i}=\sigma_{j}-1 \end{array}$$Thus, $s_{j}=s_{j+m}+1\Rightarrow s_{j}=1\wedge s_{j+m}=0$.

For counting the number of elements in the table, let $\gamma (\sigma_{i},\sigma_{j})$ be the number of elements in cell $\left(\sigma_{i},\sigma_{j}\right)$:Fact 9For $\left(\sigma_{j}^{\prime },\sigma_{j+1}^{\prime }\right)\in \left\{\left(0,0\right),\left(1,0\right),\left(0,1\right),\left(m,m\right),\left(m-1,m\right),\left(m,m-1\right)\right\}$,
(34)$$\gamma (\sigma_{j}^{\prime },\sigma_{j+1}^{\prime })=1:$$In all six cases, the histograms are from signals where either or both $S_{j}$ and $S_{j+1}$ are constant, and hence, we can trivially reconstruct the corresponding m+1 values from the histograms. We call such a histogram pair a two-trivial pair;Fact 10On the main diagonal, except $\sigma_{j}=\sigma_{j+1}=0$ and $\sigma_{j}=\sigma_{j+1}=m$,
(35)γ(σj,σj)=m−1σj+m−1σj−1:By Fact 6, $s_{j}=s_{j+m}$. For $s_{j}=0$, the possible signals for $s_{j+k},1\leq k\leq m-1$ are signals summing to $\sigma_{j}$, i.e., m−1σj, and for $s_{j}=1$, we have m−1σj−1. Since $0<\sigma_{j}<m$, therefore $\gamma (\sigma_{j},\sigma_{j})\geq 2$;Fact 11On the first diagonal above,
(36)γ(σj,σj+1)=m−1σj:By Fact 7, $s_{j}=0\wedge s_{j+m}=1$. Hence, the possible signals for $s_{j+i},1\leq i\leq m-1$ are signals summing to $\sigma_{j}$. Further, since $1<\sigma_{j+1}<m$ and $\sigma_{j+1}=\sigma_{j}+1$, therefore $0<\sigma_{j}<m-1$, and therefore, $\gamma \left(0,\sigma_{j}+1\right)=\gamma \left(m-1,\sigma_{j}+1\right)=1$ and $\gamma (\sigma_{j},\sigma_{j}+1)\geq 2$ for all other cases;Fact 12On the first diagonal below,
(37)γ(σj,σj−1)=m−1σj+1:By Fact 8, $s_{j}=1\wedge s_{j+m}=0$. Hence, the possible signals for $s_{j+i},1\leq i\leq m-1$ are signals summing to $\sigma_{j}-1=\sigma_{j+1}$. Further, since $1<\sigma_{j}<m$ and $\sigma_{j+1}=\sigma_{j}-1$, then $0<\sigma_{j+1}<m-1$, and therefore, $\gamma \left(\sigma_{j},0\right)=\gamma \left(\sigma_{j},m-1\right)=1$ and $\gamma (\sigma_{j},\sigma_{j}-1)\geq 2$ in all other cases.

For transitions, the following facts hold:Fact 13Any signal of any length n>2,m=2 can be described as a route following the arrows in the table;Fact 14An element in column *j* transitions to an element in row *j*, and as a consequence,
(38)$$\begin{array}{cc} \left(\sigma_{j},\sigma_{j}\right) & \to (\sigma_{j},\sigma_{k})\;(\mathrm{horizontal}\;\mathrm{motion}), \end{array}$$
(39)$$\begin{array}{cc} \left(\sigma_{j},\sigma_{j+1}\right) & \to (\sigma_{j+1},\sigma_{k})\;\;(\mathrm{down}\;\mathrm{motion}), \end{array}$$
(40)$$\begin{array}{cc} \left(\sigma_{j+1},\sigma_{j}\right) & \to (\sigma_{j},\sigma_{k})\;(\mathrm{up}\;\mathrm{motion}), \end{array}$$
for 0≤j−1≤k≤j+1≤m+1. Hence, only cells on the diagonal can contain intracellular paths;Fact 15Any entry is maximally m−1 steps away from a two-trivial element, since starting at element $\left[s_{0},\ldots ,s_{m-2},s_{m}\right]$, there is an m−1 path leading to $\left[s_{m-2},s_{m},\ldots ,s_{m}\right]$.

W.r.t. the number of metameric classes:Fact 16Intracellular paths for 0<i,j<m+1,|j−i|≤1, are ambiguous, since these cells contain several indistinguishable elements, and we cannot determine the path’s starting point from its histogram sequence;Fact 17Cell pairs, connected by more than one arrow in the same direction, give rise to ambiguous pairs, and paths that only contain such crossings or intracellular paths are ambiguous, since the paths cannot be distinguished by their histograms;Fact 18For m=n−1, the number of metameric classes is equal to the number of non-empty cells in the tridiagonal table minus the six two-trivial cells.
(41)κ(2,n,n−1)=n+2(n−1)−6=3n−8.For 2≤m<n−1, an upper bound on the number of metameric classes is equal to the number of ambiguous paths. We have yet to come up with a closed form for κ(n,m).

We verified the above facts by considering the transition table for m=3, as shown in Table 3. For n=m+1=4, we see that there are six uniquely identifiable signals and four metameric classes, as also confirmed by the algorithm in Table 1. For n=m+2=5, we identified two intracellular cycles in $\left(\sigma_{i},\sigma_{j}\right)=\left(1,1\right)\;\mathrm{and}\;\left(2,2\right)$. Further, there are four ambiguous cell pairs (1,1)→(1,2), (1,2)→(2,2), (2,2)→(2,1), and (2,1)→(1,1). Hence, in total, there are six metameric classes. All other arrows corresponds to non-metameric paths of which there are 18. These numbers also correspond to the results by the algorithm shown in Table 1. For n=m+3=6 and leaving out the arrows for brevity, we identified the following ambiguous paths ${\left(1,1\right)}^{3}$, ${\left(1,1\right)}^{2}\left(1,2\right)$, (1,1)(1,2)(2,2), $\left(1,2\right){\left(2,2\right)}^{2}$, and (1,2)(2,2)(2,1) and a similar set of paths starting in (2,2). Hence, we have the upper bound on the 10 metameric signals, and after a careful study, we realized that of these, there are two unique paths (1,2)(2,2)(2,1) and likewise for (2,1)(1,1)(1,2); hence, the number of metameric classes is eight, as confirmed by our algorithm; see Table 1. We have yet to identify an efficient algorithm to count all the ambiguous paths in such tables.

## 5. Conclusions

From the concept of locally orderless images [1] in image processing, we were intrigued by characterizing the metameric classes for a given set of local histograms. In this article, we took the first step by studying binary signals. We gave a sifting algorithm with a computational complexity that was factorial in the size of the window and linear in the signal size. We further identified all unique signals and an upper bound on the number of metameric classes for all signal and window sizes. While the transition tables illuminated important aspects in identifying metameric classes, we have yet to discover an efficient algorithm for this purpose. Future work includes extending our work to signals of more complex types, sets of histograms with varying window sizes, and signals of higher dimensions such as images.

## Figures and Tables

**Figure 1 entropy-24-00433-f001:**
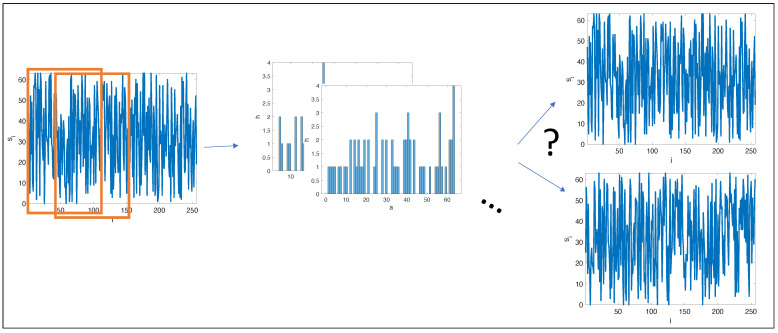
Local, overlapping histograms are calculated from a signal. Does the set of histograms uniquely identify a signal?

**Figure 2 entropy-24-00433-f002:**
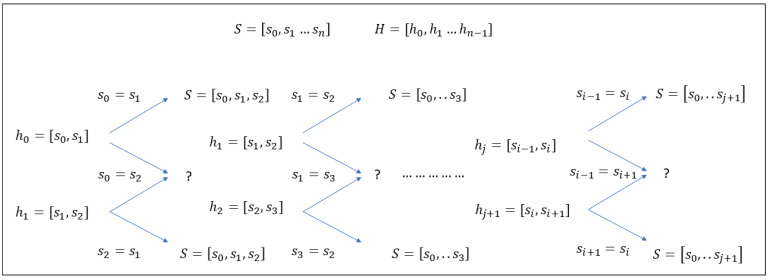
The reconstruction of the signal from a stream of local histograms. If the current point in the sequence can be resolved, the complete signal up until this point can be resolved.

**Table 1 entropy-24-00433-t001:** For different signal (*n*) and window (*m*) sizes, we calculated the total number of different signals ($2^{n}$), the number of signals not belonging to a metameric class, i.e., invertible (μ), and the number of different metameric classes (κ).

*n*	2	3	3	4	4	4	5	5	5	5	6	6	6	6	6
*m*	2	2	3	2	3	4	2	3	4	5	2	3	4	5	6
$2^{n}$	4	8	8	16	16	16	32	32	32	32	64	64	64	64	64
μ	2	6	2	14	6	2	30	18	6	2	62	46	18	6	2
κ	1	1	2	1	4	3	1	6	7	4	1	8	15	10	5

**Table 2 entropy-24-00433-t002:** All signals grouped by the sum of their local histograms when n=3 and m=2. Arrows show the relations between the $\left(\sigma_{j},\sigma_{j+1}\right)$ and $\left(\sigma_{j+1},\sigma_{j+2}\right)$ tables.

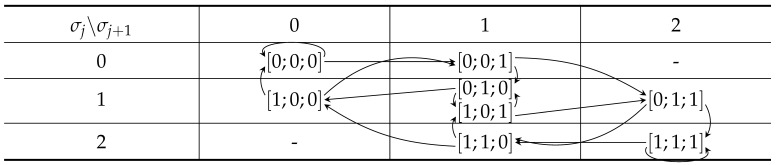

**Table 3 entropy-24-00433-t003:** All signals grouped by the sum of their local histograms when n=4 and m=3.

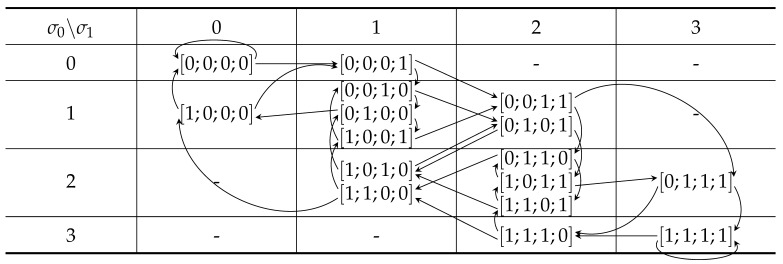

## Data Availability

Code discussed in this paper is available at https://github.com/sporring/reconstructionFromHistograms (accessed on 24 December 2021). For the specific version discussed in this paper see https://github.com/sporring/reconstructionFromHistograms/commit/333506e72fd8bea2132857869aca15b54392aa75 (accessed on 24 December 2021).

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
