# Peer review of "Reconstructing Binary Signals from Local Histograms"

_entropy, 2022, doi:10.3390/e24030433_

Round 1
Reviewer 1 Report
In this paper que authors propose provide study the reconstruction of signals based on local histograms.
The study is interesting and topical.
The article is well structured and clearly written.
The introduction could better states the article objectives and the boundaries of the proposed research. The development is interesting and is clearly presented. The authors devise an interesting framework that represents a good foundation for additional developments.
An improved discussion around non-binary signals could enrich the paper.
The conclusion can be improved by define my peremptorily stating what has been presented, what are the main novelties, advantages and disadvantages, limitation and open challenges.
It would be interesting to include:
- some clear applications for the devised formulations, in particular in what situation it would be necessary to estimate a signal or image based on a set of histograms.
- a definition of can be improved and some envisioned developments
Author Response
Color scheme: Reviewer's suggestions, author's responses
The introduction could better states the article objectives and the boundaries of the proposed research. The development is interesting and is clearly presented. The authors devise an interesting framework that represents a good foundation for additional developments.
We have clarified the introduction by adding the paragraph "A natural question, when using local histograms for signals and image analysis, is, how many signals share a given set of overlapping, local histogram (illustrated in Figure 1)? In this paper, we take a first step in answering this question
by considering densely overlapping histograms of binary signals."
An improved discussion around non-binary signals could enrich the paper.
While we very much agrees with the reviewer, that this is the goal, it is presently our focus of research, and we do not yet feel that we have substantial results that are ready for publication. Hence, we prefer to leave a discussion on non-binary signals to future research.
The conclusion can be improved by define my peremptorily stating what has been presented, what are the main novelties, advantages and disadvantages, limitation and open challenges.
We have expanded the discussion underlying the lack of an efficient algorithm for identifying ambiguous paths in the transition tables both at then end of Section 4 and in the conclusion: "While the transition tables illuminates important aspects in identifying metameric classes, we have yet to discover an efficient algorithm for this purpose."
Further challenges we believe is covered by the existing although brief statement in the conclusion: "Future work includes extending our work to signals of more complex types, sets of histograms with varying window sizes, and signals of higher dimensions such as images."
It would be interesting to include:
- some clear applications for the devised formulations, in particular in what situation it would be necessary to estimate a signal or image based on a set of histograms.
We find our contribution of purely theoretical nature, which we have underlined by adding a sentence in the introduction: "A natural question, when using local histograms for signals and image analysis, is, how many signals share a given set of overlapping, local histogram (illustrated in Figure 1)? "
Further, we have added a section on the conditional entropy of a signal giving a histogram in the introduction hinting at a coding application: "The representation power of the histogram may be quantified as the conditional entropy of signals given their histogram. For binary signals of length 3, the histogram of a signal may be summarized by its count of '1'-values, since the number of '0'-values will be 3 minus this count. For length 3 binary signals there are 8 different signals [0;0;0], [0;0;1], ... [1;1;1], which has 4 different histograms where the count of '1' values are 0, 1, 2, and 3 respectively, and the corresponding number of signals counted by their '1'-values are 1, 3, 3, and 1. Given a histogram, the conditional probability of each of these corresponding signals is thus 0, log_2 3, log_2 3, 0, and the conditional entropy may thus be found to be approximately 1.2 bits."
- a definition of [what] can be improved and some envisioned developments
Please see above.
Reviewer 2 Report
The article presents a method for reconstructing a one-dimensional signal from its local histograms that are acquired by rather short sliding windows with sliding step 1.
The paper is very logically presented (despite the reference to Equation (25) a page before its appearance), compact, yet understandable, and the writing stile is exciting and clear.
There are some grammatical errors or typos that need to be corrected, but the English language usage is good.
The tables and figures help in understanding the problem and the solution, they are well designed and aesthetic.
The itemizations could be improved, as there are three of them with different purpose, and they are referred to as 'item x' . It would be more helpful, if the facts, that the algorithm is based on (page 4) would be either referred to as 'Fact 2', or 'Fact item 2' instead of 'Item 2', or would gave a different numbering, like F1, F2, etc. The Algorithm on page 5 id rather compact, it is fine as it is, as well as the general structure of the transition tables on pages 7 and 8, but it might also help to give them a separate numbering system.
The transition tables are very nice idea to be shown, especially with the methodolically built extension from the very basic m=2, n=3 case clearly presented in the third itemization.
There are some simplifications of the method, why is see it very hardly applicable for real-life problems
A) it is for binary signals and binary histograms, and I see it very complicated to extend it to any other type of signals, whereas histograms are - according to my experience - more often used in larger alphabets
B) it considers only a single step between the sliding window (that is a bit connected to A), as this excludes the 8 bit-wise sliding) and the extension to larger steps seem to make the problem extremely complex
C) the signal is one-dimensional
D) the factorial complexity seems very much
Could the Authors very briefly comment on these considerations in their answer, how they plan to use their method in realistic signal processing problems?
The small text-errors I found
Page 2 'In [6], the authors presents' the verb is in singular
Page 3 top row 'from a regularly but sparsely set of image descriptors...' there is something missing from in front of set
Page 3, paragraph 2, line 4 there are way too many commas in the sentence after 'hence' I would remove all 3.
Page 6 after the title 4.4, there is a 'v' between a and histogram, in the 'sum of avhistogram'
In summary, the paper presents the first steps of generating a method for using histograms as metametrics in signal processing signal reconstruction, the method seems promising though hard to develop further, and the paper is very well written, the literature review is sufficient and the demonstrating tools are well selected.
Author Response
Color scheme: Reviewer's suggestions, author's respons
Thank you for your careful reading of our manuscript.
The paper is very logically presented (despite the reference to Equation (25) a page before its appearance), compact, yet understandable, and the writing stile is exciting and clear.
That has been corrected to (22).
The itemizations could be improved, as there are three of them with different purpose, and they are referred to as 'item x' . It would be more helpful, if the facts, that the algorithm is based on (page 4) would be either referred to as 'Fact 2', or 'Fact item 2' instead of 'Item 2', or would gave a different numbering, like F1, F2, etc. The Algorithm on page 5 id rather compact, it is fine as it is, as well as the general structure of the transition tables on pages 7 and 8, but it might also help to give them a separate numbering system.
Great suggestion. I've gone with 'Fact 1' and 'Step 1', and continued number of across all the fact lists.
A) it is for binary signals and binary histograms, and I see it very complicated to extend it to any other type of signals, whereas histograms are - according to my experience - more often used in larger alphabets
We are presently pursuing generalization of our results both in terms of integer signals, stride, and higher n-dimensional images, and we are not without hope, but unfortunately, our results are not yet mature enough to be included in the present submission.
B) it considers only a single step between the sliding window (that is a bit connected to A), as this excludes the 8 bit-wise sliding) and the extension to larger steps seem to make the problem extremely complex
We certainly agree that this is a very interesting extension, and we believe to be able to answer this to some extend in the near future, but as our comment to (A), our results are yet immature.
C) the signal is one-dimensional
See comment to (A)
D) the factorial complexity seems very much
Indeed, but as stated, the reconstruction algorithm in 1d is linear except for the first histogram. Hence, it is able to solve for long signals of local histograms of limited widths.
Could the Authors very briefly comment on these considerations in their answer, how they plan to use their method in realistic signal processing problems?
As a general response to A-D, Certainly, this is presently a purely theoretical contribution, and we have emphasized this somewhat by adding the following sentence to the introduction, "A natural question, when using local histograms for signals and image analysis, is, how many signals share a given set of overlapping, local histogram (illustrated in Figure 1)? Out of pure theoretical interest, in this paper we take a first step in answering this question by considering densely overlapping histograms of binary signals.".
The small text-errors I found
Page 2 'In [6], the authors presents' the verb is in singular
Page 3 top row 'from a regularly but sparsely set of image descriptors...' there is something missing from in front of set
Page 3, paragraph 2, line 4 there are way too many commas in the sentence after 'hence' I would remove all 3.
Page 6 after the title 4.4, there is a 'v' between a and histogram, in the 'sum of avhistogram'
Thank you, corrected.